# daVinci-Dev: Agent-native Mid-training for Software Engineering

**Ji Zeng** [1 2]  **Dayuan Fu** [2 3]  **Tiantian Mi** [2 3]  **Yumin Zhuang** [1 2]  **Yaxing Huang** [2 4]  **Xuefeng Li** [2 3 4]
**Lyumanshan Ye** [2 4]  **Muhang Xie** [2 3]  **Qishuo Hua** [1 2]  **Zhen Huang** [2 3]  **Mohan Jiang** [2 3 4]  **Hanning Wang** [1 2]
**Jifan Lin** [2 4]  **Yang Xiao** [2]  **Jie Sun** [2 3]  **Yunze Wu** [2 4]  **Pengfei Liu** [2 3 4]

## Abstract

While the emerging field of agentic software engineering has spurred extensive research into post-training, this paradigm alone does not fully address the distribution mismatch between traditional static pre-training and dynamic deployment environments. In this paper, we instead investigate agentic mid-training as a scalable complementary approach. Central to our approach is *agent-native data* comprising two complementary components: *contextually-native trajectories* that preserve the complete information flow an agent experiences, offering broad coverage and diversity; and *environmentally-native trajectories* whose observations stem from actual tool invocations and test executions, providing interaction authenticity. On `SWE-Bench Verified`, our recipe outperforms the previous open software engineering mid-training recipe KIMI-DEV under two post-training settings with the same base model and agentic scaffold, while using fewer than half mid-training tokens (73.1B). Furthermore, our 32B and 72B models achieve state-of-the-art resolution rates of **56.1%** and **58.5%** among open agentic recipes using agentic scaffolds, despite starting from non-coder `Qwen2.5` base models. We also observe performance gains on general code generation and scientific benchmarks. We open-source a significant portion of our datasets, recipes, and model checkpoints to facilitate further research.

[1]Zhiyuan College, Shanghai Jiao Tong University [2]Generative AI Research Lab (GAIR) [3]Shanghai Innovation Institute [4]Shanghai Jiao Tong University. Correspondence to: Pengfei Liu <pengfei@sjtu.edu.cn>.

## 1. Introduction

The capabilities of code-generating large language models have rapidly expanded from synthesizing isolated functions (Jain et al., 2025a; Wang et al., 2024b) to tackling repository-level software engineering tasks with agents (Jimenez et al., 2024), a paradigm where models iteratively navigate, edit, and test complex repositories (Jimenez et al., 2024; Badertdinov et al., 2025; Wu et al., 2026). The dominant approach to building such code agents has centered on post-training: supervised fine-tuning (SFT) on curated trajectories (Yang et al., 2025b;c) followed by reinforcement learning (RL) from execution feedback (Team et al., 2025). While effective, **the quantity and diversity are limited** for datasets that can be used in this paradigm. The process of transforming repositories into executable environments in existing works (Badertdinov et al., 2025; Yang et al., 2025b; Wang et al., 2025b) suffers from a low yield rate. Further, the yield rate of high-quality trajectories from constructed environment is constrained by the capability of existing agents and the high cost of expert human annotators, resulting in a large portion of environments unused during training (Yu et al., 2025). Such a flaw constrains the performance of SFT or RL. More fundamentally, post-training is constrained by the base model's intrinsic capacities, and certain agentic reasoning abilities may not be learnable through post-training alone (Ye et al., 2025).

This raises a natural question: *Can we instill foundational agentic behaviors earlier in the training pipeline, during mid-training?* Mid-training (MT) on domain-specific data has proven transformative for specializing LLMs to domains like mathematics (Wang et al., 2025c) and code (Yang et al., 2025c). For agentic software engineering, mid-training offers a compelling value proposition: by exposing base models to massive-scale data that mirrors agentic interactions, we can build stronger foundations that subsequent post-training can refine more efficiently. Despite this potential, *agentic mid-training remains underexplored openly*. While commercial technology reports (GLM-4.5 Team et al., 2025; Zhan et al., 2025) lack detail, existing open mid-training efforts for code models (Yang et al., 2025c) predominantly adopt a factorized approach: synthesizing isolated samples for atomic capabilities such as localization and editing, with-

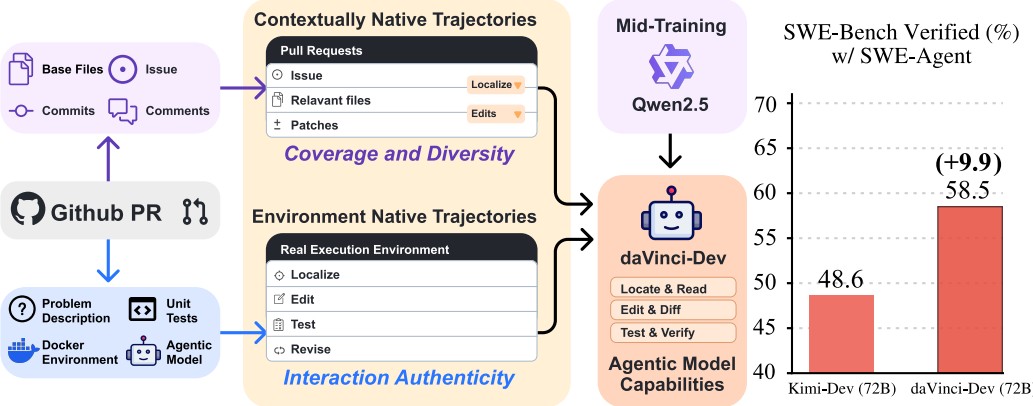

*Figure 1.* Overview of our recipe. **Left:** We curate two complementary datasets using different elements of PRs. **Right:** Comparison of the best performance of our recipe and the previous open software engineering mid-training recipe KIMI-DEV on SWE-AGENT scaffold. Detailed comparison is reported in Table 1.

out the procedural context that an agent would encounter before exercising these capabilities.

We identify the core issue in existing approaches as a **distribution mismatch** between training data and the dynamic reality of agentic deployment. To bridge this gap, we present the first systematic study of agentic mid-training for software engineering at scale. Our central thesis is that effective agentic mid-training requires **large-scale and diverse agent-native data**—supervision that preserves the complete information flow and environmental dynamics an agent experiences during deployment. We formalize this through two complementary trajectory types:

Firstly, **contextually-native trajectories** emphasizes **coverage and diversity**. Any supervision instance that preserves the structure of a realistic engineering process can be included, regardless of whether it was produced through live execution, ensuring broad source repository coverage and diversity. Supervision is organized around full task-level action sequences, bundling localization steps (e.g., identifying relevant files) together with modification steps (e.g., applying edits) with interleaved textual reasoning to reflect a coherent software development process. This allows the dataset to capture a wide variety of valid contextual patterns and operational permutations.

Secondly, **environmentally-native trajectories** prioritizes **interaction authenticity** while also considering quantity. Only trajectories generated through actual interactions with a real development environment are eligible for inclusion. These trajectories record genuine observations—tool invocations, test executions, runtime errors, and scaffold system feedback—rather than simulated or retrospectively constructed observations. We do not apply any filter strategy, so that the quantity of such trajectories can be much larger than the ones in the SFT stage. This exposes models to the dynamic feedback loops inherent in real development.

We materialize these principles through a large-scale data synthesis effort that leverages *different elements* from GitHub pull requests to construct two complementary data types, as illustrated in Figure 1. We curate a **68.6B-token contextually-native trajectories** ($\mathcal{D}^{\text{ctx}}$) using base files and commits and **3.1B-token environmentally-native trajectories** ($\mathcal{D}^{\text{env}}$) from PR-derived software engineering tasks using their Docker environments and unit tests.

Evaluating our models on `SWE-Bench Verified`, we surpass the previous state-of-the-art open MT recipe, KIMI-DEV, under two post-training settings with an aligned base model and agentic scaffold while reducing the mid-training corpus size by over 50% (73.1B vs ∼150B tokens). Our best performing 32B and 72B models reach resolution rates of **56.1%** and **58.5%**, respectively. These scores represent the highest performance among open training recipes using agentic scaffolds for their respective model sizes, a significant feat given our initialization from `Qwen2.5` (base variant) models instead of newer or code-focused base models. Beyond agentic scaffolds, this regimen also confers broad benefits, improving performance on scientific and general code generation tasks as detailed in Table 3.

To conclude, we formulate **agentic mid-training** and introduce **agent-native data** as supervision that preserves the information flow of real software engineering. We build large-scale agent-native corpora from public software development traces, including a **68.6B-token** contextually-native corpus and a **3.1B-token** set of environmentally-native rollouts, and provide a practical training recipe that leverages them. We demonstrate consistent gains on agentic software engineering brought by our agentic mid-training recipe across post-training schemes and model sizes, and provide analysis of robustness, scalability, and generalization. We release the data construction code,[1] model checkpoints, and

---

[1]Code: github.com/GAIR-NLP/daVinci-Dev. Models and

a substantial portion of curated datasets where permitted.

## 2. Background and Problem Setup

### 2.1. Agentic Software Engineering Tasks

We formalize an agentic software engineering task as a tuple $(\mathcal{R}, q, \mathcal{E})$, where $\mathcal{R}$ is a repository state, $q$ is a natural language problem description (e.g., bug report, issue), and $\mathcal{E}$ is an evaluation oracle (typically a test suite). Agentic tasks feature multi-step interaction. At each step $t$, the agent selects an action based on the conversation history and receives an observation from an observation generator:

$$a_t \sim \pi_\theta(a \mid h_{t-1}, q) \quad \text{(action selection)}$$
$$o_t \sim \text{Obs}(a_t, \mathcal{R}) \quad \text{(observation)}$$

where $h_{t-1} = \{(a_1, o_1), \ldots, (a_{t-1}, o_{t-1})\}$ accumulates prior interactions.

Actions correspond to tool calls such as searching for files, reading code, applying edits, or running tests, while observations return respective outputs. This interaction is necessary because the agent initially does not know the location of the relevant parts of the potentially large codebase, and must iteratively refine its solution based on feedback from the evaluation oracle. This complete sequence is an **agent trajectory** $\tau = (q, \mathcal{R}, \{(a_i, o_i)\}_{i=1}^T, y)$, where $y \in \{0, 1\}$ indicates whether the trajectory is successful under its supervision source.

A typical agentic software engineering workflow follows the pattern: `localize` (identifying relevant files) $\rightarrow$ `read` (understanding code context) $\rightarrow$ `edit` (applying modifications) $\rightarrow$ `test` (validating changes) $\rightarrow$ `revise` (refining based on feedback), although agents may repeat or interleave these steps as needed. This structure reflects common agent implementations (Yang et al., 2025b) and mirrors natural software development practices.

### 2.2. The Distribution Mismatch Problem

The multi-step nature of agentic software engineering contrasts with traditional training data predominantly consisting of static, completed artifacts (Figure 2a), such as outcomes—complete code files, merged commits and finished implementations. The absence of sequential action-observation pairs $(a_t, o_t)$ that agents experience at deployment creates a critical distribution mismatch: training data shows *what* was ultimately produced, but deployment requires agents to learn *how* to construct solutions through the dynamic workflow of localization, reading, editing, testing, and revision.

datasets: huggingface.co/collections/GAIR/davinci-dev.

## 3. Agent-Native Data: Design and Synthesis

To address the distribution mismatch between static training data and interactive deployment, we construct agent-native data, supervision that preserves the complete action-observation trajectories and environmental feedback agents experience during real problem-solving, which consists of two complementary types, **contextually-native trajectories**, which emphasizes coverage and diversity, and **environmentally-native trajectories**, which prioritizes interaction authenticity.

### 3.1. Contextually-Native Trajectories $\mathcal{D}^{\text{ctx}}$

#### 3.1.1. DESIGN RATIONALE

To construct the contextually-native trajectory dataset, we leverage GitHub Pull Requests (PRs) as the base data source. PRs naturally connect problem specifications (issues) to solutions (code changes) with validation signals (tests, reviews), making them well-suited for reconstructing development workflows. The key design principle is bundling complete context: rather than factorizing PRs into independent localization and editing tasks (Yang et al., 2025c) (Figure 2b), we keep all relevant information together—issue description, relevant repository files, and modifications—in a single training sample. This preserves the causal flow agents experience at deployment, where editing decisions must be conditioned on the context gathered during localization.

#### 3.1.2. CONSTRUCTION PIPELINE

**Data Sources.** We construct contextually-native trajectories from two complementary subsets: $\mathcal{D}^{\text{ctx}}_{\text{gen}}$ ("general") provides broad coverage of software engineering patterns across diverse languages and frameworks by drawing from highly-starred repositories, while $\mathcal{D}^{\text{ctx}}_{\text{py}}$ ("Python") ensures strong alignment with software engineering benchmarks (e.g., `SWE-Bench Verified`) through focused coverage of Python development. The two subsets partially overlap in Python repositories but serve complementary purposes: $\mathcal{D}^{\text{ctx}}_{\text{gen}}$ establishes cross-language understanding, while $\mathcal{D}^{\text{ctx}}_{\text{py}}$ ensures alignment with target evaluation tasks.

**Collection.** We collect pull requests through GitHub REST[2] and GraphQL APIs[3]. For each repository, we obtain pull request metadata and selectively query additional endpoints for detailed content, including linked issue descriptions (if exist), relevant file contents at the base commit, and the full commit sequence with corresponding diffs.

**Filtering.** We apply multi-level filtering criteria to ensure data quality while maintaining coverage. (1) At the repository level, $\mathcal{D}^{\text{ctx}}_{\text{gen}}$ selects from the top 10,000 most-starred

---

[2] https://docs.github.com/en/rest
[3] https://docs.github.com/en/graphql

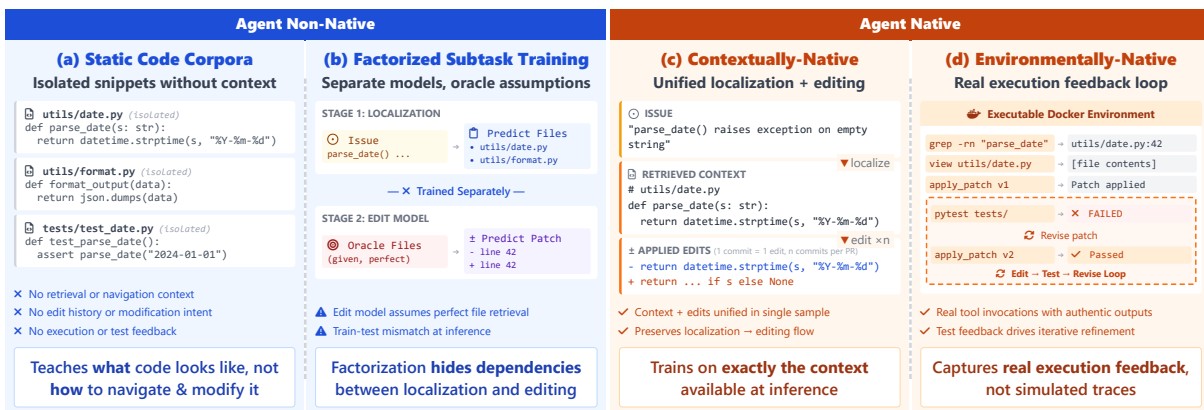

*Figure 2.* Comparison of training data paradigms. (a) Traditional code pre-training uses isolated static files. (b) Factorized approaches train subtasks separately, creating train-test mismatch. (c) Our $\mathcal{D}^{\text{ctx}}$ bundle retrieval context with sequential edit trajectory. (d) Our $\mathcal{D}^{\text{env}}$ capture real execution feedback loops.

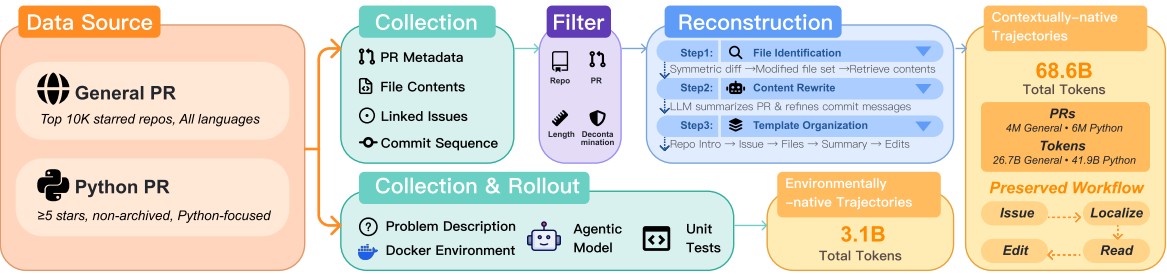

*Figure 3.* Overview of our dataset generation pipeline.

repositories across all languages. $\mathcal{D}^{\text{ctx}}_{\text{py}}$ focuses on public Python repositories with at least 5 stars and not archived. (2) At the pull request level, both subsets retain only merged and human-authored PRs. For $\mathcal{D}^{\text{ctx}}_{\text{py}}$, we additionally require modifications to be done only in Python source or documentation files, with the number of changed Python files between 1 and 5. (3) For length filtering, we discard samples exceeding 32k tokens after reconstruction. (4) For decontamination, we remove all pull requests from repositories included in `SWE-Bench Verified`.

**Reconstruction.** For each retained PR, we reconstruct an contextually-native trajectory through the following process: (1) *Relevant file identification.* We identify relevant files deterministically by querying the net diff between base and head commits. (2) *Content enhancement.* We use `Qwen3-235B-A22B-Instruct-2507` (Yang et al., 2025a) to generate two types of enhancements, including a concise PR summary that captures its intent and main changes and refined commit messages often more descriptive than the original. Detailed prompts are provided in Section C. (3) *Template organization.* We organize all extracted information into clearly delineated sections: Repository Context, Issue (when available), Pull Request, Relevant Files Found (complete file contents), PR Summary, and Edits. The Edits section contains the code modifica-

tions: For PRs with multiple commits, we concatenate them in temporal order, with each refined commit message followed by its associated code changes. The two subsets use different structural formats: $\mathcal{D}^{\text{ctx}}_{\text{gen}}$ adopts XML-like tags with traditional patch diffs and additionally includes developer comments and reviews, while $\mathcal{D}^{\text{ctx}}_{\text{py}}$ uses Markdown structure with search-and-replace blocks that more directly represent agent editing actions. See Appendix D for detailed format specifications and examples.

This organization mirrors the workflow: relevant file paths simulates the "localize" phase, file contents represent the "read" phase, edits represent the "edit" phase, and LLM-generated contents serve as textual reasoning in between.

### 3.1.3. CORPUS STATISTICS

After applying the pipeline, we obtain two complementary subsets. The general subset $\mathcal{D}^{\text{ctx}}_{\text{gen}}$ (26.7B tokens) is drawn from 4 million PRs in the $10^4$ most-starred repositories. The Python subset $\mathcal{D}^{\text{ctx}}_{\text{py}}$ (41.9B tokens) contains 6 million PRs from $7.4 \times 10^5$ repositories. Together, $\mathcal{D}^{\text{ctx}} = \mathcal{D}^{\text{ctx}}_{\text{gen}} \cup \mathcal{D}^{\text{ctx}}_{\text{py}}$ contains 68.6B tokens of contextually-native trajectories and preserves complete development workflows across diverse repositories and scenarios.

### 3.2. Environmentally-Native Trajectories $\mathcal{D}^{\text{env}}$

#### 3.2.1. DESIGN RATIONALE

While contextually-native trajectories provide agent-like *structure*, they lack agent-like *dynamics*: the model never observes the iterative feedback loop (edit $\rightarrow$ test $\rightarrow$ revise) that characterizes real agentic coding in practice. To close this gap, we curate environmentally-native trajectories—collected by running a capable agent in real executable development environments with authentic test feedback. The authenticity of our approach contrasts with trajectories in simulated or synthetic environments (Yang et al., 2025c) where codebase navigation is read-only or test execution is unavailable during rollout.

#### 3.2.2. CONSTRUCTION PIPELINE

We follow the methodology established in SWE-REBENCH (Badertdinov et al., 2025). We build a Docker image for each task that reproduces the repository state at a specific commit, alongside unit tests from the actual codebase. Then we deploy GLM-4.6 (Z.ai, 2025) within the SWE-AGENT framework (Yang et al., 2025b). For each task, we generate up to 4 rollouts, recording the complete action-observation sequences from the environment.

#### 3.2.3. CORPUS STATISTICS

After discarding trajectories exceeding 128k tokens, we obtain two types of environmentally-native trajectories: $1.85 \times 10^4$ **passing trajectories** $\mathcal{D}^{\text{env}}_{\text{pass}}$ (0.7B tokens) where all tests pass, and $5.55 \times 10^4$ **non-passing trajectories** $\mathcal{D}^{\text{env}}_{\text{fail}}$ (2.4B tokens) with test failures, totaling approximately $7.4 \times 10^4$ trajectories and 3.1B tokens. The corpus features the authentic execution feedback—test results, runtime errors and iterative refinements that complement the localization and modification patterns learned from $\mathcal{D}^{\text{ctx}}$.

## 4. Experiments

### 4.1. Training Pipeline Terminology

We clarify our position within the standard LLM development pipeline: Firstly, **pre-training** is large-scale next-token prediction on diverse corpora. Then, **mid-training** (MT) serves as an intermediate stage that shifts capability distribution by training on curated domain data at scale (Wang et al., 2025c). Unlike fine-tuning which teaches specific behaviors, mid-training operates at the knowledge level. Last, **post-training** means supervised fine-tuning (SFT) on demonstrations and/or reinforcement learning.

### 4.2. Experimental Setup

**Base model.** Unless otherwise specified, we start from Qwen2.5-72B and Qwen2.5-32B (base variant).

**Evaluation.** We evaluate on SWE-Bench Verified using SWE-AGENT (temperature 0, 128k context and 100 steps) and report Pass@1, averaged across 4 runs. Across all entries in Table 1, the largest per-entry sample standard error is $1.05\%$ (daVinci-Dev-32B); every remaining entry falls below $0.87\%$. We manually fix a small number of test cases where the provided ground truth patch cannot pass due to various reasons. The full list of adjustments is provided in Section B.

**Training stages.** We consider two stages: (i) **mid-training** (MT) on large-scale unlabeled corpora, and (ii) **supervised fine-tuning (SFT)** on agentic trajectories. The training details are in Section A.

**Data components.** For compactness in tables, we denote datasets with symbols (defined in Section 2.1). In mid-training, we study the effects of two main data components:

- **contextually-native trajectories** ($\mathcal{D}^{\text{ctx}}$): transformed GitHub pull requests into the structured format described in §3.1. the 68.6B token $\mathcal{D}^{\text{ctx}}$ is composed of $\mathcal{D}^{\text{ctx}}_{\text{py}}$ (41.9B), a Python-focused subset for alignment with software engineering benchmarks and $\mathcal{D}^{\text{ctx}}_{\text{gen}}$ (26.7B), a general subset drawn from most-starred repositories across all languages.

- **environmentally-native trajectories** ($\mathcal{D}^{\text{env}}$), collected rollouts by running SWE-AGENT with GLM-4.6 in executable Docker environments derived from real GitHub pull requests, forming $\mathcal{D}^{\text{env}}$ (3.1B raw tokens; $\sim$4.5B effective tokens). We upsample $\mathcal{D}^{\text{env}}_{\text{pass}}$ by $3\times$ during training.

For activation data, we may use:

- $\mathcal{D}^{\text{env}}_{\text{pass}}$ (0.7B): subset of $\mathcal{D}^{\text{env}}$ that pass the unit tests

- $\mathcal{D}^{\text{SWE-smith}}$ (0.11B tokens): a public set of SWE-AGENT trajectories released by Yang et al. (2025b) (mostly generated with Claude 3.7 Sonnet), which we use as an external SFT baseline.

**Baselines.** For Kimi-Dev comparisons, we quote results from Yang et al. (2025c) where applicable, and match our SFT dataset $\mathcal{D}^{\text{SWE-smith}}$ and parameters (§A) close to theirs. For experiments requiring downstream SFT on $\mathcal{D}^{\text{env}}_{\text{pass}}$ (Table 1), we utilize the official Kimi-Dev-72B checkpoint as the starting point, as their pre-RL mid-training checkpoint is not publicly available.

*Table 1.* Ablations and mid-training comparisons on `SWE-Bench Verified` (SWE-V). Our agentic mid-training on contextually-native trajectories ($\mathcal{D}^{\text{ctx}}$) and environmentally-native trajectories ($\mathcal{D}^{\text{env}}$) consistently improves downstream performance, and is competitive with or surpasses prior mid-training recipes. All results use SWE-AGENT for evaluation. [†]Trained and tested using our infrastructure. [‡]Estimated from Figure 5 in Yang et al. (2025c).

| Model / Variant | Mid-training Data | Post-training Data | Post-training Method | SWE-V |
|---|---|---|---|---|
| *Qwen 2.5 32B Series* | | | | |
| Baseline (Weak SFT)[†] | - | $\mathcal{D}^{\text{SWE-smith}}$ | SFT | 34.8 |
| Baseline (Strong SFT)[†] | - | $\mathcal{D}^{\text{env}}_{\text{pass}}$ | SFT | 53.0 |
| Ours (Weak SFT) | $\mathcal{D}^{\text{ctx}}$ | $\mathcal{D}^{\text{SWE-smith}}$ | SFT | 39.5 |
| Ours (Strong SFT) | $\mathcal{D}^{\text{ctx}}$ | $\mathcal{D}^{\text{env}}_{\text{pass}}$ | SFT | 54.1 |
| **Ours (daVinci-Dev-32B)** | $\mathcal{D}^{\text{ctx}} + \mathcal{D}^{\text{env}}$ | $\mathcal{D}^{\text{env}}_{\textbf{pass}}$ | **SFT** | **56.1** |
| *Qwen 2.5 72B Series* | | | | |
| Baseline (Weak SFT)[†] | - | $\mathcal{D}^{\text{SWE-smith}}$ | SFT | 38.0 |
| Baseline (Strong SFT)[†] | - | $\mathcal{D}^{\text{env}}_{\text{pass}}$ | SFT | 56.6 |
| Kimi-Dev (Yang et al., 2025c) | $\mathcal{D}^{\text{AgentlessMT}}$ | $\mathcal{D}^{\text{SWE-smith}}$ | SFT | $\approx 46.0^{\ddagger}$ |
| Kimi-Dev (Yang et al., 2025c) | $\mathcal{D}^{\text{AgentlessMT}}$ | $\mathcal{D}^{\text{AgentlessRL}} + \mathcal{D}^{\text{SWE-smith}}$ | SFT+RL | 48.6 |
| Kimi-Dev (Yang et al., 2025c)[†] | $\mathcal{D}^{\text{AgentlessMT}}$ | $\mathcal{D}^{\text{AgentlessRL}} + \mathcal{D}^{\text{env}}_{\text{pass}}$ | SFT+RL | 56.2 |
| Ours (Weak SFT) | $\mathcal{D}^{\text{ctx}}$ | $\mathcal{D}^{\text{SWE-smith}}$ | SFT | 46.4 |
| Ours (Strong SFT) | $\mathcal{D}^{\text{ctx}}$ | $\mathcal{D}^{\text{env}}_{\text{pass}}$ | SFT | 58.2 |
| **Ours (daVinci-Dev-72B)** | $\mathcal{D}^{\text{ctx}} + \mathcal{D}^{\text{env}}$ | $\mathcal{D}^{\text{env}}_{\text{pass}}$ | **SFT** | **58.5** |

## 4.3. Mid-Training Provides Robust Gains

For robustness we validate the effectiveness of our agent-native mid-training across two SFT regimes and against the strongest prior MT recipe (with best effort). The comparison results are shown in Table 1.

**Robustness across SFT regimes.** On the 72B model, our MT consistently boosts performance. With weak SFT, we improve from 38.0% to 46.4% with only $\mathcal{D}^{\text{ctx}}$ MT, matching the Kimi-Dev baseline despite using fewer than half the tokens (68.6B vs. 150B) and no synthetic CoT-style reasoning data or agentic rollouts in MT. With strong SFT, we reach **58.2%** with only $\mathcal{D}^{\text{ctx}}$ MT, outperforming the RL-tuned Kimi-Dev checkpoint and SFT-only baseline. This indicates that our contextually-native representation—bundling file context and edits—successfully bridges the gap between pre-training and agentic fine-tuning.

**Robustness across scales.** The benefits of our $\mathcal{D}^{\text{ctx}}$ MT recipes transfer effectively to the smaller 32B model. On the 32B scale, $\mathcal{D}^{\text{ctx}}$ MT improves the weak SFT baseline by 4.7% and the strong SFT baseline by 1.1%. This confirms that the effectiveness of our recipe is not specific to a single model capacity.

With $\mathcal{D}^{\text{ctx}} + \mathcal{D}^{\text{env}}$ MT, 72B and 32B models continue to deliver the best performance **58.5%** (+0.3%) and **56.1%** (+3.1%), respectively. Although the marginal gain on 72B falls within the per-entry standard error noted in §4.2, the larger gain at 32B (+2.0 points) and the substantial zero-shot improvements from adding $\mathcal{D}^{\text{ctx}}_{\text{py}}$ to $\mathcal{D}^{\text{env}}$ (Table 4) indicate that environmentally-native trajectories help the model internalize the dynamics of the execution environment—an effect most visible at smaller scale and in regimes where strong SFT does not already supply agentic capability.

## 4.4. Comparison with Open Recipes

We compare our full recipe against representative open methods on `SWE-Bench Verified` based on the Qwen2.5 model family and use agentic scaffolds. Table 2 presents the results. Within the 72B scale, our `daVinci-Dev-72B` achieves **58.5%**, surpassing 48.6% for Kimi-Dev using the same base model and agentic scaffold. At 32B scale, `daVinci-Dev-32B` achieves **56.1%**, which is state-of-the-art among open training recipes at this scale using agentic scaffolds, despite the fact that prior work uses `Qwen2.5-Coder-32B` series or `Qwen3-32B` while our method starts from non-coder `Qwen2.5-32B-Base`.

Beyond `SWE-Bench Verified`, we also report exploratory numbers on `SWE-Bench Pro` and `SWE-Bench Multilingual` in Section .1; because both benchmarks share repository sources with our mid-training corpus, those numbers are reported only to characterize behavior and should not be read as comparative claims.

## 4.5. Generalization Beyond SWE Tasks

While our agentic mid-training is specialized for software engineering, we investigate whether the agentic capabilities acquired from $\mathcal{D}^{\text{ctx}}$ and $\mathcal{D}^{\text{env}}$ transfer to broader domains requiring complex logic. We focus our evaluation on two distinct categories: standard code generation and rigorous scientific reasoning. In this experiment, we choose a clean single stage MT recipe with $\mathcal{D}^{\text{ctx}}_{\text{py}} + \mathcal{D}^{\text{env}}$ as dataset.

As reported in Table 3, our model demonstrates strong generalization performance, consistently surpassing the base models across both 32B and 72B scales. In code generation, we observe substantial gains on HumanEval (Chen et al., 2021) and EvalPlus (Liu et al., 2023) (after decon-

*Table 2.* Comparison with representative methods on `SWE-Bench Verified` (SWE-V). We include representative works with agentic scaffolds.

| Model | Base or Inst. | Mid-training | Post-training Method | Scaffold | SWE-V |
|---|---|---|---|---|---|
| *Qwen 2.5 32B Coder Series* | | | | | |
| R2EGym-Agent (Jain et al., 2025b) | Base | No | SFT | R2E-Gym | 34.4 |
| OpenHands-LM (OpenHands Team, 2025) | Inst. | No | SFT | OpenHands | 37.2 |
| SWE-Agent-LM (Yang et al., 2025b) | Inst. | No | SFT | SWE-Agent | 40.2 |
| SWE-Mirror-LM (Wang et al., 2025b) | Inst. | No | SFT | MOpenHands | 52.2 |
| Skywork-SWE (Zeng et al., 2025) | Inst. | No | SFT | OpenHands | 38.0 |
| SWE-Dev (Wang et al., 2025a) | Inst. | No | SFT+RL | OpenHands | 36.6 |
| *Qwen 3 32B Series* | | | | | |
| DeepSWE-Preview (Luo et al., 2025) | Inst. | No | RL | OpenHands | 42.2 |
| FrogBoss (Sonwane et al., 2025) | Inst. | No | SFT | SWE Agent | 54.6 |
| SWE-Lego-Qwen3-32B (Tao et al., 2026) | Inst. | No | SFT | OpenHands | 52.6 |
| *Qwen 2.5 32B Series* | | | | | |
| **daVinci-Dev-32B (Ours)** | **Base** | **Yes** | **SFT** | **SWE-Agent** | **56.1** |
| *Qwen 2.5 72B Series* | | | | | |
| Kimi-Dev (Yang et al., 2025c) | Base | Yes | SFT+RL | SWE-Agent | 48.6 |
| **daVinci-Dev-72B (Ours)** | **Base** | **Yes** | **SFT** | **SWE-Agent** | **58.5** |

*Table 3.* Generalization performance on scientific and code benchmarks. We report the base model performance and the impact of our MT stages. **MT** refers to the model trained on $\mathcal{D}^{\mathrm{ctx}}_{\mathrm{py}} + \mathcal{D}^{\mathrm{env}}$.

| | Qwen2.5-32B | | | Qwen2.5-72B | | |
|---|---|---|---|---|---|---|
| Benchmark | Base | MT | Δ | Base | MT | Δ |
| *Scientific Benchmarks* | | | | | | |
| GPQA-Main | 38.17 | 38.84 | +0.67 | 43.30 | 44.87 | +1.57 |
| SuperGPQA | 33.85 | 35.94 | +2.09 | 37.76 | 39.27 | +1.51 |
| SciBench | 18.46 | 20.49 | +2.03 | 19.33 | 19.77 | +0.44 |
| *Code Benchmarks* | | | | | | |
| HumanEval | 58.16 | 81.42 | +23.26 | 64.27 | 76.73 | +12.46 |
| EvalPlus | 50.13 | 71.31 | +21.18 | 56.04 | 69.45 | +13.41 |
| DS-1000 | 12.20 | 21.20 | +9.00 | 21.40 | 24.70 | +3.30 |

*Table 4.* Ablation of data components in MT. **Top:** In a zero-shot setting (no SFT), grounding $\mathcal{D}^{\mathrm{env}}$ with $\mathcal{D}^{\mathrm{ctx}}$ yields massive gains (+7.7% on 72B). **Bottom:** Even when performing SFT, exposing the model to $\mathcal{D}^{\mathrm{env}}$ during MT improves final performance (comparing rows 3 and 4).

| | | | SWE-Verified | |
|---|---|---|---|---|
| MT Data | Tokens | SFT Data | 32B Base | 72B Base |
| *Ablation: $\mathcal{D}^{\mathrm{ctx}}$ vs. $\mathcal{D}^{\mathrm{env}}$ (Zero-shot / No SFT)* | | | | |
| $\mathcal{D}^{\mathrm{env}}$ | 4.5B | – | 43.7 | 47.1 |
| $\mathcal{D}^{\mathrm{env}} + \mathcal{D}^{\mathrm{ctx}}_{\mathrm{py}}$ | 46.4B | – | **49.9** | **54.8** |
| *Impact of MT Composition on SFT* | | | | |
| $\mathcal{D}^{\mathrm{ctx}}_{\mathrm{py}}$ | 41.9B | $\mathcal{D}^{\mathrm{env}}_{\mathrm{pass}}$ | 52.9 | 56.5 |
| $\mathcal{D}^{\mathrm{env}} + \mathcal{D}^{\mathrm{ctx}}_{\mathrm{py}}$ | 46.4B | $\mathcal{D}^{\mathrm{env}}_{\mathrm{pass}}$ | **53.6** | **57.8** |
| $\mathcal{D}^{\mathrm{env}} + \mathcal{D}^{\mathrm{ctx}}$ | 73.1B | $\mathcal{D}^{\mathrm{env}}_{\mathrm{pass}}$ | 56.1 | 58.5 |

tamination following `XCoder` (Wang et al., 2024b)), confirming that our data improves fundamental coding proficiency. More notably, we observe transfer learning to scientific benchmarks such as GPQA (Rein et al., 2024) and SciBench (Wang et al., 2024a). These tasks, which demand expert-level domain knowledge and multi-step reasoning, benefit from the decision-making patterns inherent in our agentic mid-training. This suggests that the logic required for autonomous software engineering fosters fundamental reasoning skills that generalize beyond code.

# 5. Analysis

In this section, we analyze the factors contributing to the effectiveness of agentic mid-training. We first examine the efficiency and information density of agent-native data, then explore the synergistic relationship between our two data types, and finally discuss the scalability of this paradigm.

## 5.1. High Information Density and Efficiency

A key advantage of our approach is token efficiency. Kimi-Dev's recipe involves 70B tokens directly derived from PR plus 20B synthetic trajectory/CoT tokens upsampled

4 times, totaling ~**150B tokens**. In contrast, our **68.6B tokens** $\mathcal{D}^{\mathrm{ctx}}$ MT stage consistently outperforms Kimi-Dev as shown in section 4.3, and performance further grows with additional **4.5B effective tokens** $\mathcal{D}^{\mathrm{env}}$ added to MT training stage. This efficiency stems from our $\mathcal{D}^{\mathrm{ctx}}$ being closer to software engineering agent's test distribution compared to factorized approaches, and our $\mathcal{D}^{\mathrm{env}}$ being more authentic than simulated trajectories.

## 5.2. The Complementary Nature of $\mathcal{D}^{\mathrm{ctx}}$ and $\mathcal{D}^{\mathrm{env}}$

While both $\mathcal{D}^{\mathrm{ctx}}$ and $\mathcal{D}^{\mathrm{env}}$ exhibit gains when used in isolation, their effects complement each other when combined. Table 4 presents an ablation study on the composition of MT data across both 32B and 72B model scales.

$\mathcal{D}^{\mathrm{env}}$ **require** $\mathcal{D}^{\mathrm{ctx}}$ **grounding.** In the zero-shot setting (top section), training on environmentally-native trajectories alone yields 47.1% (72B). However, mixing in the Python contextually-native subset $\mathcal{D}^{\mathrm{ctx}}_{\mathrm{py}}$ boosts performance to 54.8%—a significant **+7.7%** gain. This suggests that while environmentally-native trajectories teach the model *how* to interact with the environment, contextually-native

data provides the necessary *knowledge* and code modification diversity required to solve complex issues.

**Mid-training on $\mathcal{D}^{\mathrm{env}}$ aids SFT.** A key question in agent training is whether "double-dipping"—training on $\mathcal{D}^{\mathrm{env}}$ during MT and then fine-tuning on them during SFT—provides value. Comparing the first two rows of the SFT section (Table 4), we observe a consistent improvement when $\mathcal{D}^{\mathrm{env}}$ are included in MT. For the 72B model, adding $\mathcal{D}^{\mathrm{env}}$ to the MT mix improves the final SFT score from 56.5% to **57.8%**. This indicates that mid-training, with no loss mask applied, allows the model to internalize the dynamics of the execution environment more deeply than SFT alone. Finally, our strongest result, **58.5%** (72B) and **56.1%** (32B) comes from scaling the contextually-native foundation from $\mathcal{D}^{\mathrm{ctx}}_{\mathrm{py}}$ (41.9B) to the full $\mathcal{D}^{\mathrm{ctx}}$ (68.6B), showing that the sheer scale and diversity of contextually-native supervision remain the dominant factors in model performance.

## 5.3. Behavioral Effects of Mid-Training

Beyond aggregate resolution rates, we examine *how* mid-training changes the agent's behavior on `SWE-Bench Verified`. We extract behavioral indicators from agent trajectories across all 8 experimental settings in Table 1 (2 model sizes × 2 SFT regimes × {base, mid-trained}), each repeated 4 times. We restrict reporting to indicators that achieve high signal-to-noise ratio (SNR) across the 4 repeated runs; other indicators we examined—error-recovery patterns, planning structure, debugging-cycle counts—did not reach sufficient cross-run stability to be reported reliably.

We focus on four localization-related indicators computed from the agent's file-level actions:

- **Loc. Recall (Edit)**: fraction of gold-patch files the agent edited.

- **Loc. Precision (Edit)**: fraction of agent-edited files appearing in the gold patch.

- **View→Edit Ratio**: fraction of uniquely viewed files that were also edited.

- **Unique Files Viewed**: number of distinct files the agent opened.

Table 5 reports the resulting values. Under the Weak SFT regime, mid-training delivers a clear localization improvement: relative to the no-MT baseline, edit-level recall and precision rise by +10.5 pp and +10.0 pp at 32B (SNR ≈ 29 for both) and by +5.2 pp and +3.2 pp at 72B (SNR ≈ 7.6 and 2.2), with the sign of each gain consistent across all 4 runs per setting. The number of unique files viewed is essentially unchanged, indicating that mid-training improves the model's ability to identify *which* files to act on rather than its

*Table 5.* Behavioral indicators averaged across 4 runs per setting on `SWE-Bench Verified`. "MT" denotes the mid-trained model paired with $\mathcal{D}^{\mathrm{ctx}}$ under Weak SFT and with $\mathcal{D}^{\mathrm{ctx}} + \mathcal{D}^{\mathrm{env}}$ under Strong SFT.

| | Qwen2.5-32B | | | | Qwen2.5-72B | | | |
| | Weak SFT | | Strong SFT | | Weak SFT | | Strong SFT | |
| Indicator | Base | MT | Base | MT | Base | MT | Base | MT |
| --- | --- | --- | --- | --- | --- | --- | --- | --- |
| Loc. Recall (Edit) | 0.627 | 0.732 | 0.779 | 0.790 | 0.735 | 0.787 | 0.797 | 0.798 |
| Loc. Precision (Edit) | 0.450 | 0.550 | 0.205 | 0.213 | 0.549 | 0.581 | 0.217 | 0.205 |
| View→Edit Ratio | 0.490 | 0.569 | 0.383 | 0.407 | 0.556 | 0.577 | 0.399 | 0.412 |
| Unique Files Viewed | 2.38 | 2.36 | 3.65 | 3.49 | 2.52 | 2.55 | 3.45 | 3.47 |

exploration breadth. Consistent with this, the View→Edit Ratio rises by +7.9 pp at 32B Weak SFT (SNR ≈ 10) and trends upward in the other 7 settings, indicating the agent commits to a higher fraction of the files it inspects. Under the Strong SFT regime, the no-MT baseline already attains high localization recall (~ 0.78–0.80), leaving little room for further improvement; the positive trend is largely preserved but the absolute deltas shrink. The localization gains from mid-training are qualitatively consistent across 32B and 72B, with larger absolute improvements at 32B where the base model has more headroom.

## 5.4. Scalability: from raw PRs to executable tasks

Our approach is scalable along two axes: data availability and empirical performance scaling.

**Empirical Scaling.** Figure 4 illustrates the learning curves of both `Qwen2.5-72B` and `Qwen2.5-32B` during the MT stage on the $\mathcal{D}^{\mathrm{ctx}}_{\mathrm{py}} + \mathcal{D}^{\mathrm{env}}$ mixture. We observe a robust log-linear relationship between training steps and Pass@1 performance for both model sizes ($R^2 \approx 0.90$ and $0.89$ respectively), suggesting that performance has not saturated. This indicates that further scaling would yield continued improvements.

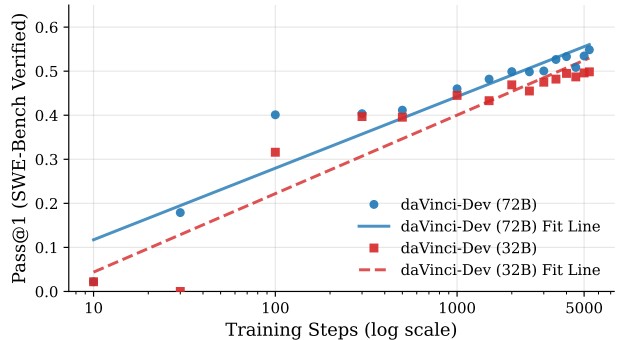

*Figure 4.* **Scaling Law of Agent-Native Mid-training.** Pass@1 performance on `SWE-Bench Verified` during mid-training (MT) on the $\mathcal{D}^{\mathrm{ctx}}_{\mathrm{py}} + \mathcal{D}^{\mathrm{env}}$ mixture. The strong log-linear fit indicates that agentic capabilities scale predictably with training steps and data consumption, suggesting the model has not yet saturated.

**Scaling** $\mathcal{D}^{\text{ctx}}$**.** The Python-focused subset $\mathcal{D}^{\text{ctx}}_{\text{py}}$ is built from $\sim 1.3 \times 10^7$ PRs (before filtering) in $\sim 7.4 \times 10^5$ repositories while the multi-language subset $\mathcal{D}^{\text{ctx}}_{\text{gen}}$ only utilize the $1 \times 10^4$ most starred repositories. However, our survey indicates there are $\sim 3 \times 10^8$ PRs in $\sim 10^9$ public repositories, suggesting strong scaling potential for the corpus by expanding language coverage and relaxing filters.

**Scaling** $\mathcal{D}^{\text{env}}$**.** Recent advances in environment construction (Badertdinov et al., 2025) demonstrate that raw PRs can be automatically transformed into *deeper* supervision: executable tasks with Docker environments and unit tests, constructed via a fully automated pipeline. Such tasks are the source of $\mathcal{D}^{\text{env}}$. Importantly, SWE-REBENCH builds its public dataset from **3,468** Python repositories and reports **21,336** validated tasks, suggesting substantial headroom for scaling as repository coverage and yield rate expand.

## 6. Related Work

**Mid-training** Recent work increasingly positions mid-training as a critical bridge between large-scale pre-training and post-training. Rather than transitioning directly from noisy, web-scale corpora to SFT or RL, mid-training introduces higher-quality, task-structured, or instruction-oriented data at later stages of training, often paired with learning-rate annealing (Mo et al., 2025; Zhang et al., 2025; Tu et al., 2025). For example, OctoThinker (Wang et al., 2025c) argues that mid-training can substantially improve both the sample efficiency and the achievable performance ceiling of subsequent RL by stabilizing internal representations and encouraging reasoning-friendly behaviors. Despite these advances, public detail on agentic mid-training data and pipelines remains limited. Kimi-Dev (Yang et al., 2025c) incorporates supervision for behaviors such as file retrieval and file editing during mid-training, but treats these behaviors in isolation rather than as part of a coherent, end-to-end agentic process. More recent system reports such as KAT-Coder (Zhan et al., 2025) and GLM-4.5 (GLM-4.5 Team et al., 2025) also describe leveraging agentic or PR-derived signals during mid-training; however, neither documents the data construction process in comparable detail nor releases the underlying mid-training corpora. In contrast, this paper presents a transparent and reproducible pipeline for agent-native mid-training data, treats agentic behavior as a first-class training objective with complete trajectories, and releases the construction methodology together with a substantial portion of the resulting datasets.

**Agentic training** Agentic training builds upon prior work, SFT and RL. Early agents were predominantly trained by applying SFT on small models using trajectories generated by closed-source large models in specific environments (Zeng et al., 2024; Chen et al., 2024; Xi et al., 2025; Fu et al., 2025b). Others use these ideas to create datasets in the domain of code agents (Yang et al., 2025b; Sonwane et al., 2025; Badertdinov et al., 2025; Wang et al., 2025b; Guo et al., 2025). With the introduction of GRPO (Shao et al., 2024; DeepSeek-AI et al., 2025), recent work has increasingly focused on training agents capable of multi-step reasoning and tool usage with reinforcement learning (Zheng et al., 2025; Li et al., 2025; Team et al., 2025). Despite a growing body of research on agentic post-training, systematic studies of agentic mid-training remain notably scarce. Since mid-training can utilize more diverse data than the post-training stage, exploring its role and potential benefits becomes important.

**Data synthesis** Early approaches to synthetic data primarily focused on recombining and rewriting large-scale corpora and using reject-sampling to get the final data (Yuan et al., 2023). As the demand for data scale and coverage increased, persona-driven synthesis was introduced (Ge et al., 2025; Fu et al., 2025a), enabling a systematic expansion of the task space beyond naturally occurring data. More recently, the synthesis of agentic processes emerged. Through interaction within synthetic environments (DeepSeek-AI et al., 2025; Team et al., 2025; Badertdinov et al., 2025), models actively generate data containing decision-making trajectories, feedback loops, and long-horizon dependencies. In this paper, we focus on mid-training synthesis, serving as a critical bridge between pre-training and agentic post-training.

## 7. Conclusion

We introduced agentic mid-training to bridge the distribution gap between static pre-training and dynamic agent deployment. By curating 71.7B tokens of *agent-native data*—unifying contextually diverse information flow with environmentally authentic interactions—our recipe outperforms the leading open mid-training baseline on `SWE-Bench Verified` while using less than half the data. We achieve the highest resolution rates among open agentic recipes using agentic scaffolds, 56.1% and 58.5% with our 32B and 72B model despite starting from non-coder `Qwen-2.5`, demonstrating that instilling agentic behaviors mid-training yields robust, scalable gains. We open-source our code, datasets (where permitted) and checkpoints to accelerate research into natively agentic foundation models.

## Impact Statement

This paper presents work aimed at advancing the field of Machine Learning, specifically within the domain of automated software engineering. Our goal is to improve the reliability and efficiency of AI agents that assist developers, potentially reducing the burden of routine bug fixes and maintenance.

We recognize the ethical and legal implications of training large language models on public code repositories. To address intellectual property and licensing concerns, we have implemented strict protocols for the open-source portion of our datasets. Our curation process for contextually-native trajectories and environmentally-native trajectories prioritizes repositories with permissive licenses. The released dataset retains original license metadata to ensure attribution is possible. Furthermore, we have imposed access controls on the dataset repository that require users to explicitly agree to adhere to the terms of the original source licenses before accessing the data.

Regarding the model checkpoints, we acknowledge that code generation models can inadvertently reproduce patterns or snippets from their training data. In the documentation accompanying our model releases, we mandate that users verify the licensing requirements of any generated code and utilize appropriate security scanning and human review before deployment.

While our work aims to increase productivity, we also acknowledge standard risks associated with code-generating AI, such as the potential generation of insecure code or dual-use scenarios. We emphasize that these agents are designed for human-in-the-loop workflows rather than autonomous deployment without oversight.

## Acknowledgments

We express our gratitude to Haoyang Zou, Zengzhi Wang, and Fan Zhou for their constructive feedback and stimulating discussions. We are also grateful to Liming Liu for his guidance and advice.

This work was partially funded by the National Natural Science Foundation of China (62476168) and SJTU School of Electronic Information and Electrical Engineering – ByteDance LLM Joint Laboratory.

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

### .1. Behavior on Broader SWE Benchmarks

**Caveat: training-data contamination.** We additionally evaluate `daVinci-Dev` on the public split of `SWE-Bench Pro` (Deng et al., 2025) and `SWE-Bench Multilingual` (Yang et al., 2025b) to characterize behavior beyond the Python-centric `SWE-Bench Verified` setting. Both benchmark splits are derived from public GitHub repositories that overlap heavily with the sources of our mid-training corpus: their pull requests, gold patches, and substantial portions of the corresponding repository snapshots appear verbatim in $\mathcal{D}^{\text{ctx}}$. We therefore do *not* claim relative superiority over other models on these benchmarks, and we caution against cross-paper comparison. We report these numbers only to describe how our models behave on broader task distributions than the one targeted by our training pipeline.

**Results.** Under the public splits and the same SWE-AGENT scaffold used for `SWE-Bench Verified` (temperature 0, 128k context, 100 steps), our models reach the resolution rates in Table 6. Scaling from 32B to 72B is consistently positive on both benchmarks; absolute numbers remain well below `SWE-Bench Verified`, reflecting the harder and less Python-centric task distributions.

*Table 6.* Resolution rates (%) on `SWE-Bench Pro` (public split) and `SWE-Bench Multilingual` under SWE-AGENT. Both benchmarks share repository sources with our mid-training corpus; numbers should be read in light of the contamination caveat above.

| Model | SWE-Bench Pro | SWE-Bench Multilingual |
|---|---|---|
| daVinci-Dev-32B | 11.1 | 16.7 |
| daVinci-Dev-72B | 16.4 | 23.3 |

## A. Training Details

### A.1. Dataset Components and Staging

**PR MT Staging.** Our $\mathcal{D}^{\text{ctx}}$ (68.6B) training was conducted in two sequential stages rather than a single mix. We first trained on the general subset $\mathcal{D}^{\text{ctx}}_{\text{gen}}$ (26.7B tokens) to establish a broad software engineering baseline. We then performed mid-training (MT) on the Python subset $\mathcal{D}^{\text{ctx}}_{\text{py}}$ (41.9B tokens) to specialize the model on agent-native, Python-centric patterns. Our $\mathcal{D}^{\text{ctx}} + \mathcal{D}^{\text{env}}$ (73.1B) was also conducted in two sequential stages where the first stage is the general subset $\mathcal{D}^{\text{ctx}}_{\text{gen}}$ (26.7B tokens) and the second stage is the other two datasets.

**SFT Configuration.** For all SFT experiments involving our $\mathcal{D}^{\text{env}}_{\text{pass}}$ or $\mathcal{D}^{\text{SWE-smith}}$ datasets, we trained for **5 epochs**.

### A.2. Hyperparameters

We provide the key hyperparameters used for Mid-training (MT) and Supervised Fine-tuning (SFT) below.

**MT Hyperparameters.** We use a global batch size of 1024 samples and a peak learning rate of $8 \times 10^{-5}$. The learning rate schedule utilizes a warmup ratio of 0.05 (5% of total training steps), followed by cosine decay until all samples are consumed once (1 epoch). No loss mask is applied during MT.

**SFT Hyperparameters.** We use a global batch size of 128 samples and a peak learning rate of $1 \times 10^{-5}$. The learning rate schedule utilizes a warmup ratio of 0.10 (10% of total training steps), followed by cosine decay until all samples are consumed once per epoch. A standard loss mask is applied to user and tool tokens during SFT.

**Shared optimizer settings.** Both the MT and SFT stages share the same optimizer configuration. We use Adam with $\beta_1 = 0.9$, $\beta_2 = 0.95$, weight decay 0.1, cosine learning-rate decay, and zero attention/hidden dropout. Gradient clipping is applied with threshold 1.0. These values follow the defaults in the public `THUDM/slime` training script,[4] which we use as our training backend.

---

[4] https://github.com/THUDM/slime/blob/main/scripts/run-qwen3-4B-base-sft.sh

## B. Evaluation-Infrastructure Fixes for SWE-Bench Verified

For reproducibility, this appendix enumerates the evaluation-infrastructure adjustments we apply when running `SWE-Bench Verified`. These adjustments target the harness, not the benchmark tasks themselves and not any model-generated patch. They are intended to reduce false negatives on instances where the public gold patch is already known to fail under the default harness, and to make the benchmark runnable in the serverless Kubernetes environment we use. Many of the underlying issues are documented in the SWE-Bench issue tracker, and the corresponding issue/PR numbers are noted inline. After these adjustments, the relative ranking across models is unchanged across all evaluations we run in this paper. The same harness is used for every model reported in the main text, so all comparisons within the paper are made under a single, consistent evaluation setup.

- **`astropy__astropy-7606` — stale `PASS_TO_PASS`.** A `PASS_TO_PASS` entry remained in the Verified split although it had already been removed from the main SWE-Bench dataset (issues #223, #267, #484). We ignore this stale entry during validation.

- **`astropy__astropy-8707` and `astropy__astropy-8872` — dependency mismatch.** Both instances fail gold-patch validation due to package-version incompatibilities (#267, #484). We update the NumPy-related dependencies in the per-instance environment so that the gold patch executes.

- **`django__django-10097` — path resolution under large batches.** The gold patch fails only under highly parallel batched execution; running the affected `PASS_TO_PASS` case in isolation passes, indicating a path-resolution race rather than a patch-correctness issue (#267, #294, #487). We evaluate this case with `--parallel 1`, consistent with the current upstream harness, which treats Django specially.

- **`psf__requests-*` — external `httpbin` instability.** Several `psf__requests-*` instances depend on the external `httpbin` service, whose availability has been documented as a source of false negatives (#484). We deploy a local mirror that reproduces the official service behavior and route evaluation through it.

- **`sphinx-doc__sphinx-*` — pytest output parser.** A pytest-version-dependent output-format mismatch causes a one-line pytest summary (e.g., "`1 passed, 0 failed, 0 errors`") to be misparsed as a failure (#228). We force `pytest -rA` so that case-level outcomes are emitted, consistent with later upstream fixes.

- **`sphinx-doc__sphinx-*` — patch reset.** Pre-existing diffs in setup files such as `tox.ini` can cause `patch` to execute in reverse during evaluation, and reset logic can accidentally restore the entire repository to `base_commit` when `test_patch` contains only new files (PR #453, PR #475, #518). We preserve required setup modifications across reset.

- **Region-specific network failures.** A small number of instances and normal software dependent workflow depend on external websites unreachable from our network region (such as `pypi.org` or `github.com`); we reverse proxy the required external endpoints so development and testing can run normally.

## C. LLM prompts used for PR rendering

During context enrichment (Section 3.1), we optionally call an LLM to (i) generate a concise pull-request summary and (ii) normalize/optimize commit messages for readability. We use `Qwen3-235B-A22b-Instruct-2507` with fixed output budgets (512 tokens for PR summaries; 256 tokens for commit-message refinement).

## D. Dataset Formats

We include the templates for two types of data in $\mathcal{D}^{\text{ctx}}$: (i) the **General PR** format in $\mathcal{D}^{\text{ctx}}_{\text{gen}}$, and (ii) the **Python PR** format in $\mathcal{D}^{\text{ctx}}_{\text{py}}$.

The General PR format uses XML-like tags similar to `The Stack v2` (Lozhkov et al., 2024) and includes rich interaction history (comments and reviews). Events related to a pull requests are concatenated in chronological order. Different from `The Stack v2`, we always include relevant file content and grouped review comments threads. We also insert LLM-generated action summaries as chain of thoughts. This corpus is sourced from top-starred repositories without the 1–5 Python-file constraint. An example is shown in Figure 6.

**PR Summary Prompt**

```
Summarize this pull request in 1-4 clear sentences:

Repository: {{.RepoName}}
Description: {{.RepoDesc}}

PR Title: {{.Title}}
PR Description:
{{.Body}}

{{if .Issue}}Related Issue: {{.Issue.Title}}
{{.Issue.Body}}

{{end}}Changed Files:
{{range .ChangedPyFiles}}- {{.}}
{{end}}

Commits:
{{range .Commits}}
## Message: {{.Message}}

Changes:
{{range .Diffs}}
File: {{.Path}}
{{.Patch}}
{{end}}
{{end}}

Please provide a clear and concise summary (1-4 sentences) of this Pull Request, focusing on:
1. What problem does it solve or what feature does it add?
2. What are the key changes made?
3. Any important implementation details?

Summary:
```

**Commit Message Refinement Prompt**

```
Optimize this commit message for clarity and educational value while keeping it concise.

PR Context Summary: {{.Summary}}

Original commit message:
{{.Commit.Message}}

Diff Context:
{{range .Commit.Diffs}}File: {{.Path}}
{{truncatePatch .Patch 2000}}

{{end}}
Provide an optimized version that:
1. The subject is clear and descriptive
2. If the commit is trivial and the changes are minimal, don't add the footer
3. Otherwise, keep the footer in one sentence

Refined commit message:
```

*Figure 5.* Prompts used for optional LLM-based PR summary generation and commit-message refinement during PR rendering.

---

**General PR Example**

**# Repository Context**
Name: parcel-bundler/parcel
Description: The zero configuration build tool for the web.
**# Relevant Files Context**
**## packages/core/parcel-bundler/src/cli.js**

```
...
  if (command.name() === 'serve' && command.target === 'browser') {
    const server = await bundler.serve(
      command.port || 1234,
      command.https,
      command.host
    );
...
```

**Response:**
<pr>Title: use env port
Adds `process.env.PORT` as a default port option...
<chain_of_thought>I'll start by... </chain_of_thought>
<pr_comment>I think it should rather be...
<pr_review>Looks good to me.
<pr_review_state>approved
<pr_commit>use env port
<commit_file>packages/core/parcel-bundler/src/cli.js
<patch>

```
@@ -219,7 +219,7 @@ async function bundle(main, command) {
    if (command.name() === 'serve' && command.target === 'browser') {
      const server = await bundler.serve(
-       command.port || 1234,
+       process.env.PORT || 1234,
        command.https,
        command.host
    );
```

</patch>
<pr>
<pr_status>closed
<pr_is_merged>True

---

*Figure 6.* Example of the General PR format in $\mathcal{D}_{\text{gen}}^{\text{ctx}}$.

The Python PR format uses a Markdown structure and represents edits in a *search-and-replace* action space. It includes an LLM-generated PR summary after presenting all related files (simulating a overall planning and reasoning phase in an agentic workflow) and enhanced commit messages (simulating textual reasoning before action). Edits are rewritten from git diff format to search-replace format used in many agentic scaffolds. The **# Issue** section is omitted if no linked issue is found. An example is shown in Figure 7.

---

**Python PR Example**

# Repository Context
Name: Pylons/waitress
Description: Waitress - A WSGI server for Python 3
# Issue
## \xa0 and \x85 are stripped from header values
Given that these bytes are allowed in header values (due to `obs-text`), they shouldn't be stripped during header-field OWS stripping...
# Pull Request
## Bugfix: Don't strip whitespace from values before inserting into environ
This fixes a small bug where the value of the header would get stripped when inserted into the environ so it no longer matched. Closes #432
# Relevant Files Found
## src/waitress/task.py

```
...
        for key, value in dict(request.headers).items():
            value = value.strip()
            mykey = rename_headers.get(key, None)
...
```

# Edits
This pull request removes the erroneous `.strip()` call on header values in the WSGI environ construction. The HTTP specification allows certain non-ASCII bytes (`\xa0`, `\x85`) in header values via `obs-text`, and these should not be stripped.
Remove the strip() call from header value processing in get_environment()
Edit: src/waitress/task.py
Search:

```
        for key, value in dict(request.headers).items():
            value = value.strip()
            mykey = rename_headers.get(key, None)
            if mykey is None:
                mykey = "HTTP_" + key
```

Replace:

```
        for key, value in dict(request.headers).items():
            mykey = rename_headers.get(key, None)
            if mykey is None:
                mykey = "HTTP_" + key
```

---

*Figure 7.* Example of the Python PR format in $\mathcal{D}_{\text{py}}^{\text{ctx}}$.

# E. Benchmark decontamination

In our training dataset we take measures to remove samples related to the `SWE-Bench Verified` benchmark as detailed in Section 3.1. For the HumanEval and EvalPlus benchmarks, we follow the decontamination procedure of `XCoder` (Wang et al., 2024b). Concretely, for each benchmark instance we form the reference text by concatenating the prompt and canonical solution, tokenize it, and compute the set of unique $n$-grams ($n = 13$). We then scan the tokenized training corpus and, for every training sample, compute its set of unique 13-grams and the overlap with each benchmark instance. Similarity is measured as a leakage ratio:

$$\text{leakage\_ratio}(e, x) = \frac{|G_e \cap G_x|}{|G_e|},$$

where $G_e$ is the set of unique 13-grams in the benchmark instance and $G_x$ is the set of unique 13-grams in a training sample. For each benchmark instance, we take the maximum leakage ratio over all training samples as its contamination score.

We manually selected the contamination threshold as $\tau = 0.10$ based on case studies of high-overlap matches. Using this criterion, we identified 24 contaminated HumanEval instances, which were removed from evaluation.

