# OpenReview forum: "daVinci-Dev: Agent-native Mid-training for Software Engineering"
_ICML.cc/2026/Conference — ICML 2026 spotlight_

### Official Review · Reviewer_hTYp · 2026-03-11

**Soundness:** 4
**Presentation:** 3
**Significance:** 4
**Originality:** 3
**Overall Recommendation:** 5
**Confidence:** 3

**Summary:**

This paper introduces the paradigm of agent-native mid-training and presents daVinci-Dev, an training recipe and model suite backed by a newly constructed large-scale, agent-native mid-training dataset. The authors experimentally validate the core motivation behind their data design: effective agentic mid-training requires complete, end-to-end trajectories rather than factorized sub-trajectories split by task. Furthermore, the paper demonstrates the necessity of combining contextually-native trajectories (which encapsulate the problem, intermediate reasoning, ground-truth locations, and ground-truth edits) with environmentally-native trajectories (which capture the dynamic, trial-and-error iterative editing processes of an actual agent in a real execution environment).

**Compliance With Llm Reviewing Policy:**

Affirmed.

**Final Justification:**

The authors have addressed my concern, and I will raise my score to 5.

**Key Questions For Authors:**

Please see weakness above.

**Limitations:**

yes

**Strengths And Weaknesses:**

Strengths :
1. The paper clearly defines the requisite paradigm for constructing mid-training data tailored for agents, specifically addressing the distribution mismatch through agent-native mid-training.
2. It provides a high-quality, large-scale dataset encompassing both contextual and environmental trajectories and open-sources a significant portion of it to the community.
3. The authors exhibit a highly clear design logic backed by comprehensive ablation studies that effectively validate the necessity and complementary nature of the two proposed trajectory types.
4. The work offers valuable empirical insights into scaling laws and the effectiveness of double-dipping.

Weaknesses:
1. The environmentally-native trajectories are generated using GLM-4.6, but the paper does not discuss whether the resulting models can reach or approach the performance level of this teacher model on the evaluated benchmarks.
2. The paper claims high training efficiency based on lower token counts compared to prior work , but it overlooks the massive computational overhead required to construct the dataset, such as deploying a 235B parameter model across millions of pull requests and running GLM-4.6 for trajectory rollouts.
3. The paper lacks qualitative analysis and case studies demonstrating the actual behavioral shifts of the model. It remains unclear whether the quantitative performance gains stem from better localization logic, superior code editing, or more robust error recovery strategies.

---

> ### Author Rebuttal · Authors · 2026-03-31
>
> Thank you for your insightful comment!
>
> W1: Thank you for raising this point. Using the same SWE-Agent scaffold and evaluation setup, GLM-4.6 reaches **62.0%** on SWE-Bench Verified, while our resulting models reach **56.1%** (32B) and **58.5%** (72B). In other words, the gap to the teacher is **5.9 points** for daVinci-Dev-32B and **3.5 points** for daVinci-Dev-72B. We therefore do approach the teacher closely, especially at 72B, but we do not fully match it.
>
> W2: We agree that dataset construction has a nontrivial upfront cost, and lower training-token count should not be interpreted as meaning the entire pipeline is cheap end-to-end. Our efficiency claim is narrower: it is about the token efficiency of the resulting training recipe relative to prior mid-training approaches.
>
> At the same time, the construction cost is largely a one-time investment that is amortized across many downstream training runs, ablations, and model scales. In our case, the same constructed corpus has already supported multiple experiments in the paper and can continue to be reused in future runs, whereas training cost is paid again for every model. Releasing a large portion of the dataset also helps amortize this cost at the community level rather than keeping it private. We will clarify this distinction in the revised paper, and if space permits we will add a more explicit discussion of dataset-construction overhead as a practical limitation.
>
> W3: We analyze behavioral indicators extracted from agent trajectories across all 8 experiments (2 model sizes × 2 SFT schemes × base/mid-trained. The mid-training data paired with the Weak SFT scheme is `D_ctx` while the Strong SFT is paired with `D_ctx + D_env`), each repeated 4 times. We focus on indicators with high signal-to-noise ratio (SNR) across the 4 repeated runs; other behavioral metrics (e.g., error recovery patterns, planning structure, debugging cycles) did not achieve sufficient cross-run stability to report reliably.
>
> We measure four localization-related indicators from the agent's file-level actions:
>
> - **Loc. Recall (Edit)**: fraction of gold-patch files that the agent edited
> - **Loc. Precision (Edit)**: fraction of agent-edited files that appear in the gold patch
> - **View→Edit Ratio**: fraction of uniquely viewed files that were also edited
> - **Unique Files Viewed**: number of distinct files the agent opened
>
> | Indicator | 32B base Weak | 32B MT Weak | 32B base Strong | 32B MT Strong | 72B base Weak | 72B MT Weak | 72B base Strong | 72B MT Strong |
> |---|---|---|---|---|---|---|---|---|
> | Loc. Recall (Edit) | 0.627 | **0.732** | 0.779 | 0.790 | 0.735 | **0.787** | 0.797 | 0.798 |
> | Loc. Precision (Edit) | 0.450 | **0.550** | 0.205 | 0.213 | 0.549 | **0.581** | 0.217 | 0.205 |
> | View→Edit Ratio | 0.490 | 0.569 | 0.383 | 0.407 | 0.556 | 0.577 | 0.399 | 0.412 |
> | Unique Files Viewed | 2.38 | 2.36 | 3.65 | 3.49 | 2.52 | 2.55 | 3.45 | 3.47 |
>
> The clearest behavioral shift from mid-training is improved **localization accuracy under the Weak SFT scheme**: the mid-trained model edits the correct files at substantially higher rates (+10.5% recall, +10.0% precision at 32B; +5.2%, +3.2% at 72B), with consistent sign across all 4 runs per experiment (SNR > 7 at 32B, > 2 at 72B). This gain occurs without viewing more files — the number of unique files viewed is unchanged — indicating that mid-training improves the model's ability to identify which files to act on, not its exploration breadth.
>
> Under the Strong SFT scheme, the base model already achieves high localization recall (~0.79–0.80), leaving less room for improvement, but the positive trend mostly remain.
>
> Regarding model size: the localization gains from mid-training are qualitatively consistent across 32B and 72B, with larger absolute improvements at 32B where the base model has more room to improve. We do not observe obviously different behavioral patterns between the two sizes from the indicators with sufficient cross-run stability.

---

> > ### Author Rebuttal · Reviewer_hTYp · 2026-04-02
> >
> > The authors have addressed my concern, and I will raise my score to 5.

---

> > > ### Author Response · Authors · 2026-04-07
> > >
> > > Thank you so much for your positive feedback and for raising your score to a 5! We are very glad that our clarifications and the new behavioral analysis fully addressed your concerns.
> > >
> > > Your initial questions prompted us to dig deeper into the quantitative behavioral shifts (such as localization accuracy without mere exploration breadth), which has significantly strengthened the analytical depth of our work. We will be sure to include this analysis, as well as the discussions regarding teacher-model performance gaps and dataset construction overhead, in the final manuscript. Thank you again for your excellent suggestions!

---

### Official Review · Reviewer_Gzks · 2026-03-12

**Soundness:** 3
**Presentation:** 3
**Significance:** 4
**Originality:** 3
**Overall Recommendation:** 5
**Confidence:** 4

**Summary:**

This work introduces agent-native mid-training, a paradigm that treats agentic behavior as a primary training objective by leveraging diverse GitHub PR elements. The paper provides a systematic analysis of how contextual coverage and environmental authenticity synergistically improve agent performance across different model scales.

**Compliance With Llm Reviewing Policy:**

Affirmed.

**Final Justification:**

My final recommendation is Accept. The paper introduces a highly original and significant approach to agent-native mid-training, supported by clear scaling laws and authentic interaction data. The paper's core methodology is sound, the presentation is clear.

The authors' rebuttal successfully addressed my primary concerns.

**Key Questions For Authors:**

- Rationale Behind Format Choices and Generalization. Why does the Python subset adopt a specific Markdown format while the general subset uses XML? Does this alignment with the evaluation scaffold limit the model's generalization to agents with different action spaces?
- Impact of RL on Mid-Training Gains. How does this mid-training framework interact with subsequent Reinforcement Learning (RL), and is there a risk that the MT gains might be overridden by a strong RL phase?
- Scalability to Stronger Base Models. The experiments show impressive gains on Qwen2.5 Base variants. Would these relative margins of improvement decrease if the recipe were applied to newer models that already incorporate extensive agentic training, such as the Qwen3 series?
- Justification for Benchmark Modifications. Regarding the manual fixes to SWE-Bench Verified test cases, could you detail exactly what was changed and justify why these modifications do not compromise the integrity of the comparison?

**Limitations:**

yes

**Strengths And Weaknesses:**

Strengths
- In-depth Data Synergy Analysis: The ablation studies convincingly demonstrate that $\mathcal{D}^{env}$ requires $\mathcal{D}^{ctx}$ grounding for maximum efficacy, providing clear guidance for future agentic data composition.
- Predictable Scaling of Agentic Capabilities: The authors establish a robust log-linear scaling law ($R^2 \approx 0.90$) between training steps and Pass@1 performance, indicating high potential for further scaling.
- Authentic Interaction Dynamics: By incorporating real execution feedback and runtime errors instead of simulated traces, the model internalizes the dynamic feedback loops necessary for autonomous SE tasks.

Weaknesses
- Distillation from Strong Agentic Models. The reliance on GLM-4.6 to generate environment trajectories ($\mathcal{D}^{env}$) introduces potential distillation from a model that already possesses strong agentic capabilities. This raises questions about whether the gains stem from raw environmental interaction or from the distilled "teacher" agent's strategies.
- Contradictory Claims on Reasoning Data Synthesis. The claim of using no synthetic reasoning data in Section 4.3 contradicts the methodology of using Qwen3-235B for PR summaries and chain-of-thought insertion. This inconsistency obscures the true impact of the raw PR traces versus the LLM-generated enhancements.

---

> ### Author Rebuttal · Authors · 2026-03-31
>
> Thank you for your insightful comment!
>
> W1: Our experiments where the mid-training data only involves `D_ctx` also showed visible gains over those without mid-training, and the gain persists across model sizes and SFT regimes (Table 1, Section 5.1). For example, on the 72B model, `D_ctx`-only MT improves the weak SFT baseline from 38.0% to 46.4% and the strong SFT baseline from 56.6% to 58.2%; similar gains appear at 32B (34.8→39.5% weak, 53.0→54.1% strong). Notably, our 68.6B-token `D_ctx` MT stage consistently outperforms Kimi-Dev's 150B-token MT recipe under comparable post-training (46.4% vs. ≈46.0% with weak SFT), achieving this with fewer than half the tokens and no synthetic reasoning data. This shows the gains from the raw interaction structure in our data.
>
> W2: Please see Reviewer 488r Q3 for clarification on this issue.
>
> Q1: The short answer is that this was a pragmatic design choice rather than the result of a controlled format ablation. The general subset was built earlier and uses an XML-like structure because it is convenient for preserving richer metadata such as PR comments, reviews, and patch structure across diverse repositories. The Python subset was constructed later with a stronger focus on agent-style editing, and Markdown with search-and-replace blocks felt more natural and fluent for representing those trajectories.
>
> We do not claim that Markdown itself is inherently superior to XML, and we did not run a controlled comparison between the two formats because of training cost. Our view is that the main learning signal comes from the bundled problem-context-edit structure, not from the surface delimiter syntax. We therefore do not think this choice fundamentally ties the model to one specific evaluation scaffold or action space. In fact, this mixed-format design is consistent with later PR-derived training recipes as well: for example, Qwen3-Coder-Next (Sec. 3.1.1) explicitly represents edits using both Search-and-Replace and standard git diff formats within the same PR-based data pipeline. Our setup is similar in spirit: the Python subset uses a search-and-replace style, while the general XML-like subset uses git-diff-style edits. If anything, exposure to both renderings should make the model less brittle to any single interface.
>
> Q2: We did not study the interaction between our mid-training recipe and a subsequent RL phase in this work, so we do not want to overclaim here. At the moment, we view this as an important open question beyond the scope of this paper.
>
> Q3: Due to time and resource constraints we mid-trained `Qwen3-14B` (based variant) on about `2/3` of `D_ctx_gen` or about `1/6` of our total MT data. We then fine-tuned the resulting model on `D_env_pass`. The resulting model reached 37.9% on SWE Bench Verified. For comparison, the same base model directly SFT-ed on `D_env_pass` without mid-training reached 37.0%. The relative gains are visible despite the small mid-trained subset we used. All MT and SFT hyperparameters are aligned with appendix A.
>
> Q4: Please see Reviewer 488r Q1 for the detailed modification list.
>
> We believe these modifications do not compromise the relative performance comparison because:
>
> - **Relative, not absolute metrics:** Our focus is the relative improvement of our recipe over baselines. If modifications affect both daVinci-Dev and baselines similarly, the ranking and comparative gains remain valid.
> - **Infrastructure consistency:** All models evaluated in this paper use the same evaluation infrastructure, ensuring fair comparison within our experiments.
>
> for community trust and reproducibility, we will provide the detailed fix list in the camera-ready version.

---

> > ### Author Rebuttal · Reviewer_Gzks · 2026-04-03
> >
> > Thank you for the detailed rebuttal. I will raise my overall score to 5.

---

> > > ### Author Response · Authors · 2026-04-07
> > >
> > > Thank you so much for your review, your engaging questions, and for choosing to raise your overall score to a 5! We are very glad that our rebuttal resolved your concerns.
> > >
> > > Your insights regarding our data formatting, interaction dynamics, and benchmark modifications were incredibly helpful. We will absolutely ensure that the final camera-ready version reflects these detailed discussions, including the explicit list of evaluation-infrastructure fixes. Thank you again for your time and support!

---

### Official Review · Reviewer_488r · 2026-03-13

**Soundness:** 3
**Presentation:** 3
**Significance:** 4
**Originality:** 3
**Overall Recommendation:** 4
**Confidence:** 4

**Summary:**

The paper presents daVinci-Dev, a token-efficient mid-training recipe for software engineering agents using 71.7B tokens of agent-native data. By unifying contextually-native and environmentally-native trajectories, daVinci-Dev achieves state-of-the-art results on SWE-Bench Verified with half less data than previous approaches, demonstrating that agentic capabilities can be effectively and efficiently instilled before the post-training stage. The authors plan to opensource the datasets, recipes, and model chekcpoints, contributing to the community for future research.

**Compliance With Llm Reviewing Policy:**

Affirmed.

**Final Justification:**

The author's rebuttal fully address my concern.

**Key Questions For Authors:**

Q1. The authors mention that they manually fix a small number of test cases in SWE-Bench Verified. Could you provide a detailed list of these modifications and report the scores on the strictly unmodified benchmark to ensure fair comparison?

Q2.  Could you provide the missing training hyperparameters, such as weight decay and specific optimizer settings, omitted from Appendix A for full reproducibility?

Q3. Please clarify the claim "no synthetic reasoning data" as pointed out in the weakenesses.

**Limitations:**

yes

**Strengths And Weaknesses:**

## Strengths ##

- **Remarkable Token Efficiency**: The proposed recipe outperforms the KIMI-DEV baseline while using less than half the mid-training tokens (73.1B vs ~150B), demonstrating a highly optimized data-scaling approach.

- **Superior Performance from Base Models**: The model achieves SOTA resolution rates despite being initialized from non-coder Qwen2.5-Base variants, which highlights the intrinsic value of the curated agent-native corpora.

- **Strong Domain Generalization**: Beyond software engineering, the mid-training recipe provides significant gains in scientific reasoning (GPQA) and standard code generation (HumanEval), suggesting the acquisition of fundamental reasoning capabilities.

- **Strong Domain Generalization**: Beyond software engineering, the mid-training recipe provides significant gains in scientific reasoning (GPQA) and standard code generation (HumanEval), suggesting the acquisition of fundamental reasoning capabilities.

- **Community Contribution**: The datasets, recipes, and model chekcpoints, if opensourced, will significantly help the community on agentic mid-training.


## Weaknesses ##
- **Overclaimed Novelty** In line 40, the claim that "agentic mid-training remains underexplored" overlooks recent Agentic-CPT efforts such as KAT-Coder[1] and GLM-4.5[2]. The authors should properly contextualize their novelty against the literature to better reflect the contribution.

- **Contradictory Claims on Synthetic Reasoning Data** Line 264 explicitly claims the use of "no synthetic reasoning data",  which contradicts the fact that Qwen3-235B was used to generate PR summaries and textual reasoning. This inconsistency makes it difficult to disentangle the benefits of the raw data structure from the distilled high-quality reasoning.

[1] KAT-Coder Technical Report. https://arxiv.org/abs/2510.18779

[2] GLM-4.5: Agentic, Reasoning, and Coding (ARC) Foundation Models. https://arxiv.org/abs/2508.06471

---

> ### Author Rebuttal · Authors · 2026-03-31
>
> Thank you for your insightful comment!
>
> W1: You are right that our framing should better acknowledge recent agentic mid-training work such as KAT-Coder and GLM-4.5. We will revise the wording accordingly. Our intended novelty claim is not that prior work never used agentic or PR-derived mid-training data, but that we provide a detailed and reproducible construction pipeline for agent-native mid-training data, together with a large-scale empirical study and a planned public release of a substantial portion of the dataset. In particular, while the works you mention also appear to use PR-derived data, they do not describe the data construction process in comparable detail, and their training data is not publicly available. We will therefore soften the novelty claim and emphasize transparency, reproducibility, and openness as the key distinction.
>
> Q1: These fixes were evaluation-infrastructure fixes rather than modifications to benchmark tasks or model-generated patches, and many of them correspond to issues already discussed publicly in the SWE-Bench community. Concretely, they include: a stale `PASS_TO_PASS` entry for `astropy__astropy-7606`; dependency/environment compatibility fixes for `astropy__astropy-8707` and `astropy__astropy-8872`; path-resolution instability for `django__django-10097` under large batched evaluation; replacing unstable external `httpbin` dependencies for several `psf__requests-*` instances with a local mirror; and parser / patch-reset issues affecting a subset of `sphinx-doc__sphinx-*` cases. We also adapted the runtime from the original local Docker setup to the serverless k8s environment we use, and resolved region-specific network failures that otherwise make some instances unevaluable from our location.
>
> A strictly unmodified score from the local public SWE-Bench harness is therefore not available from our setup, because some instances cannot be run reliably without those runtime and network fixes. However, after only migrating the runtime and fixing connectivity, without changing any individual harness logic, our scores are consistently 1-2 points below the official `sb-cli` service, and the relative ranking between models does not change. We will provide the full instance list and a clearer breakdown of these fixes in the camera-ready version so the community can map our reported scores to the underlying evaluation setup.
>
> Q2: The shared optimizer settings for our mid-training and supervised fine-tuning are: Adam with `beta1=0.9`, `beta2=0.95`, `weight_decay=0.1`, cosine decay, and zero attention / hidden dropout (all of them are the default values in https://github.com/THUDM/slime/blob/main/scripts/run-qwen3-4B-base-sft.sh). The training backend also uses gradient clipping with the default threshold `1.0`. We will add these exact settings to Appendix A so the training recipe is fully reproducible.
>
> Q3: We acknowledge the potential confusion. We will revise the wording to say "no synthetic solution-style reasoning data or synthetic agent rollouts" rather than implying there is no model-generated text at all. To clarify the distinction:
>
> By "no synthetic reasoning data," we mean that we do not use either of the two synthetic components described in Kimi-Dev Appendix A. First, Kimi-Dev constructs **synthetic reasoning data** in a one-shot style: a model reasons extensively about a bug-fixing or test-writing problem and directly produces a patch or test-case solution, without relying on tool calls or iterative agentic loops. Second, Kimi-Dev constructs **synthetic agentic interactions**, where the model is rolled out with a synthetic, execution-free tool interface that mimics parts of the agent workflow. Our `D_ctx` dataset does not contain either type of supervision. It does contain model-generated PR summaries and refined commit messages, but these are lightweight metadata enrichments layered on top of real PR records rather than synthesized solutions or interaction traces. They are meant to better expose the likely intent behind observed developer actions, but they are still substantially less synthetic than generating one-off reasoning solutions, localization traces, editing decisions, or full tool-use trajectories.
>
> We will clarify this distinction in the camera-ready version to avoid future confusion.

---

> > ### Author Rebuttal · Reviewer_488r · 2026-04-03
> >
> > My concerns are fully addressed. I will keep positive about the paper.

---

> > > ### Author Response · Authors · 2026-04-07
> > >
> > > Thank you very much for your time, your positive assessment, and your valuable suggestions! We are thrilled that our rebuttal fully addressed your concerns.
> > >
> > > We will make sure to incorporate all the clarifications discussed—including the contextualization of related work, the exact training hyperparameters, the detailed list of runtime/network fixes for SWE-Bench Verified, and the more precise wording regarding synthetic reasoning data—into the final camera-ready version. Thank you again for your constructive guidance!

---

### Official Review · Reviewer_yMNi · 2026-03-16

**Soundness:** 3
**Presentation:** 3
**Significance:** 4
**Originality:** 3
**Overall Recommendation:** 5
**Confidence:** 4

**Summary:**

The paper proposes a training approach for improving large language models used as software engineering agents. The authors argue that standard LLM training creates a mismatch between static pretraining on text/code and the interactive environments where coding agents operate. To address this, they introduce agent-native mid-training, a stage inserted between pretraining and post-training that exposes the model to trajectories resembling real software-engineering workflows. These trajectories come from two sources: reconstructed GitHub pull-request histories that capture issue context and code modifications, and agent interaction traces collected by running coding agents in executable repositories. Training on such trajectories aims to teach models how to reason over repositories, make edits, and interact with tools. Experiments show that models trained with this mid-training stage achieve improved performance on SWE-Bench Verified and other coding tasks compared to baselines without agent-native mid-training.

**Compliance With Llm Reviewing Policy:**

Affirmed.

**Final Justification:**

I maintain my strong prior assessment. The paper achieves strong SOTA performance within their model class and provides insight into the training process through several ablations and studies. The clarifications in the rebuttal will help the readability of the paper.

**Key Questions For Authors:**

- Do you make any effort to ensure that your dataset isn’t contaminated with SWE-bench verified repos?
- What qualitative agent behaviors actually improve? (planning, tool use, debugging?, code writing)
- Does mid-training help different sized models in different ways?
- when you mix D_ctx vs D_env, how do you sample from the two datasets during training?
- How does the final model perform on SWE-bench Pro? It would be nice to include this or some other extensions of SWE-bench verified (e.g. multilingual)
- what is the difference between weak and strong sft?
- how important is the post-training in your experiments? How well do models perform without post-training? It’s difficult to separate the impact of mid and post training from the experiment results since Table 1 uses inconsistent forms of post-training
- Kimi-Dev post trained on D_env_pass achieves 56.2 on swe-bench verified. THis makes it seem like post training on D_env_pass is the most important part of the pipeline compared to mid training.

**Limitations:**

- swe-bench verified is starting to become outdated. Inclusion  (even just for one version of the final model)  of swe-bench pro or multilingual would help bring the results up to date and contextualize them.
- limited and inconsistent ablations make it difficult to understand which parts of the pipeline are most critical to the performance

**Strengths And Weaknesses:**

Strengths:
- The paper achieves strong SOTA performance within their model classes
- A significant portion of the dataset and model is to be open-sourced
- the paper explores transfer learning gains beyond coding
- The emphasis on mid-training is a nice shift from recent works that focus on post-training for SWE-agents

Weaknesses
- the impact of environmentally-native trajectories is poorly explained and supported. The experiments involving environmentally-native trajectories only show a 0.3% (72B) and 2.0% (32B) gain
- the paper makes no attempt at quantifying the variance in their results across different rollout seeds or training runs.
- the paper has limited ablations so it’s difficult to conclude which aspect of the pipeline (agent interactions, the specific PR reconstruction dataset, training on more tokens) leads to the performance gains
- it’s not clear whether the main benefits of the training data are found in mid training or post training

---

> ### Author Rebuttal · Authors · 2026-03-31
>
> Thank you for your insightful comment!
>
> W1 & W2: We appreciate this scrutiny. All SWE-V results are averaged over 4 runs; the largest sample standard error across all experiments is 1.05% (daVinci-Dev-32B), with the rest of entries below 0.87%.
>
> We acknowledge that the 0.3% gain on 72B from adding `D_env` in Table 1 (58.2→58.5) falls within this noise margin and cannot be claimed as statistically significant in isolation. However, we believe `D_env`'s contribution is better assessed through the full pattern of evidence across settings and scales:
>
> - **Zero-shot (Table 4, no SFT):** Adding `D_ctx_py` to `D_env` yields +6.2% (32B) and +7.7% (72B), far exceeding noise. This is `D_env`'s clearest value: it enables agentic capability that `D_ctx` alone cannot provide without SFT.
> - **SFT ablation (Table 4):** On 72B, including `D_env` in MT improves the SFT result from 56.5→57.8 (+1.3%), exceeding the typical SE.
> - **32B scale (Table 1):** The gain is +2.0% (54.1→56.1), also above noise.
>
> We agree with the reviewer that `D_ctx` is the dominant factor in our recipe. `D_env`'s role is complementary — it helps the model internalize execution dynamics — and its impact is most visible at 32B scale and in the zero-shot regime. We will revise the paper to state these conclusions more precisely and avoid overclaiming the 72B marginal gain.
>
> W3: We have conducted ablations addressing these concerns: Sec. 5.2 (Table 4) ablates the contribution of agent interactions (`D_env`) and the specific PR reconstruction dataset (`D_ctx`), showing their individual and combined effects across both zero-shot and SFT settings. Sec. 5.3 (Figure 4) examines the effect of training on more tokens, revealing a log-linear scaling relationship between training steps and Pass@1 performance that has not yet saturated.
>
> W4: The experiments in Sec. 5.2 and 5.3 directly show that mid-training alone provides substantial gains. In Table 4 (zero-shot, no SFT), `D_env` mid-training alone reaches 43.7%/47.1% for 32B/72B, and `D_env + D_ctx_py` reaches 49.9%/54.8%. These results demonstrate that mid-training contributes significant agentic capability independently, before any post-training is applied.
>
> Q1: Yes, we removed all pull requests from repositories included in SWE-Bench
> Verified.
>
> Q2: Please see Reviewer hTYp W3 for a detailed behavioral analysis.
>
> Q3: Please see Reviewer hTYp W3 for the model-size comparison.
>
> Q4: We mix `D_ctx` and `D_env` through **simple concatenation and shuffling**, ensuring each sample is seen exactly once during mid-training.
>
> Q5: We evaluated our final models on SWE Bench Pro (public split) and SWE Bench Multilingual on modal platform.
>
> | Model | Pro | Multilingual |
> |---|---|---|
> | daVinci-Dev 32B | 11.1% | 16.7% |
> | daVinci-Dev 72B | 16.4% | 23.3% |
>
> It's worth noting that both benchmarks have nearly 100% contamination with our mid-training data, in a sense that the pull requests, gold patches and some updated files of the source repositories are seen literally during mid-training. Therefore, these results should be interpreted with caution and we do not claim relative superiority over other models on these benchmarks.
>
> Q6: The distinction between "weak SFT" and "strong SFT" refers to the **post-training dataset source and quality**:
>
> | Aspect | Weak SFT | Strong SFT |
> |--------|----------|-----------|
> | **Post-training Data** | `D_swe` (0.11B tokens) | `D_env_pass` (0.7B tokens) |
> | **Data Source** | External SWE-Agent trajectories from prior work (mostly Claude 3.7 Sonnet) | Passing trajectories from `D_env` (collected via GLM-4.6 rollouts in real environments) |
>
> Our mid-training further improves both regimes compared to baseline that does not use mid-training, demonstrating that `D_ctx` mid-training provides orthogonal benefits to post-training dataset quality.
>
> Q7: See Table 4 for the raw performance of the mid-trained models without post-training. In particular, with no SFT at all, `D_env` mid-training alone reaches 43.7% / 47.1% on SWE-Bench Verified for the 32B / 72B models, and `D_env + D_ctx_py` reaches 49.9% / 54.8%. This shows that mid-training already contributes substantial agentic capability before post-training, while post-training remains important for pushing the final scores to 56.1% / 58.5%.
>
> Q8: These questions highlight the need to disentangle mid-training gains from post-training gains. We address this through ablations in **Table 1**.
>
> Even on the strongest baseline (56.6% strong SFT), our `D_ctx` mid-training delivers **+1.6% gain** on the 72B model to reach 58.2%, and the full recipe (`D_ctx + D_env`) further improves to **58.5%**.
>
> This ablation directly answers the concern: mid-training is neither redundant nor merely riding on strong post-training, but rather establishes a stronger foundation that post-training can further refine.

---

> > ### Author Rebuttal · Reviewer_yMNi · 2026-03-31
> >
> > I appreciate you answering my questions and would like to see these clarifications included in the final paper.
> >
> > In relation to the new eval on SWE-bench Pro, it's unfortunate that there is contamination, but I think their inclusion in the paper would be nice with the same contamination disclaimer.
> >
> > Overall, my concerns are addresses and I maintain my strong rating.

---

> > > ### Author Response · Authors · 2026-04-07
> > >
> > > Thank you for your continued support and for your highly constructive feedback throughout the review process! We are glad that our rebuttal fully addressed your concerns.
> > >
> > > We completely agree with your suggestion regarding SWE-bench Pro and Multilingual. We will ensure that these evaluations, along with the explicit contamination disclaimer, are included in the final camera-ready version to provide full context to the community. We will also incorporate the other clarifications discussed in the rebuttal. Thank you again for helping us strengthen the paper!

---

### Decision · Program_Chairs · 2026-04-30

**Decision:**

Accept (spotlight)

**Comment:**

This paper makes a strong contribution on agent-native mid-training for software engineering agents, and the reviewers were broadly positive about both its empirical strength and its potential value to the community. In particular, they found the core idea well motivated, the results strong within the evaluated model class, and the analysis of contextual and environmental trajectories useful for understanding what kinds of agentic data matter. The rebuttal addressed the main concerns by clarifying the role of different data sources, adding behavioral analysis, reporting variance, and making the claims around novelty and synthetic reasoning data more precise. Overall, the paper is technically solid, timely, and worthy of acceptance.